

# Improving Consistency in Methane Emission Quantification from the Natural Gas Distribution System across Measurement Devices

Judith Tettenborn[1], Daniel Zavala-Araiza[1,2], Daan Stroeken[1], Hossein Maazallahi[1*], Carina van der Veen[1], Arjan Hensen[3], Ilona Velzeboer[3], Pim van den Bulk[3], Felix Vogel[4], Lawson Gillespie[4,5], Sebastien Ars[4], James France[6,7], David Lowry[6], Rebecca Fisher[6], and Thomas Röckmann[1]

[1]Institute for Marine and Atmospheric Research Utrecht (IMAU), Utrecht University, Utrecht, The Netherlands
[2]Environmental Defense Fund, Amsterdam, The Netherlands
[3]Netherlands Organisation for Applied Scientific Research (TNO), Utrecht, The Netherlands
[4]Climate Chemistry Measurements and Research, Climate Research Division, Environment and Climate Change Canada, Toronto, Canada
[5]Department of Physics, University of Toronto, Toronto, Canada
[6]Department of Earth Sciences, Centre of Climate, Ocean and Atmosphere, Royal Holloway, University of London, Egham, United Kingdom
[7]Environmental Defense Fund, London, United Kingdom
[*]Now at: Department of Renewable Energies and Environment, College of Interdisciplinary Science and Technologies, University of Tehran, Tehran, Iran

**Correspondence:** Thomas Röckmann (t.roeckmann@uu.nl)

**Abstract.**

Mobile real-time measurements of ambient methane provide a fast and effective method to identify and quantify methane leaks from local gas distribution systems in urban areas. The objectives of these methodologies are to i) identify leak locations for repair and ii) construct measurement-based emission rate estimates, which can improve emissions reporting and contribute to monitoring emission changes over time. Currently, the most common method for emission quantification uses the maximum methane enhancement detected while crossing a methane plume. However, the recorded maximum depends on instrument characteristics, such as measurement cell size, pump speed and measurement frequency. Consequently, the current approach can only be used by instruments with similar characteristics. We suggest that the integrated spatial peak area is a more suitable quantity that can eliminate the bias between different instruments. Based on controlled release experiments conducted in four cities (London, Toronto, Rotterdam, and Utrecht), emission estimation methodologies were evaluated. Indeed, the integrated spatial peak area was found to be a more robust metric across different methane gas analyzer devices than the maximum methane enhancement. A statistical function based on integrated spatial peak area is proposed for more consistent emission estimations when using different instruments. On top of this systematic relation between actual emission rate and recorded spatial peak area, large variations in methane spatial peak area were observed for the multiple transects across the same release point, in line with previous experiments. This variability is the main contributor of uncertainty in efforts to use mobile measurements to prioritize leak repair. We show that repeated transects can reduce this uncertainty and improve the categorization into different leak categories. We recommend a minimum of three and an optimal range of 5-7 plume transects for effective emission quantification to prioritize repair actions.



## 1 Introduction

Mitigating methane ($CH_4$) emissions is an important step to combat climate change. The increase in atmospheric $CH_4$ has contributed approximately 30% of global warming since pre-industrial times (IPCC (2023)). $CH_4$ has an 82 times higher global warming potential over 20 years (Smith et al. (2021)), and a shorter atmospheric lifetime ($9.1 \pm 0.9$ years (Szopa et al. (2021))) than $CO_2$. This offers the opportunity for reductions in climate warming by reducing $CH_4$ emissions, possibly avoiding temperature overshoots (Collins et al. (2018), Hu et al. (2013)). General awareness of the importance for timely $CH_4$ mitigation is rising. More than 150 nations have joined the Global Methane Pledge established during the UN Climate Change Convention Of Parties (COP 26) in 2021, pledging to reduce human-made $CH_4$ emissions in 2030 by 30% relative to 2020 (EU Commission (2021)). Following that, regulations specifically targeting $CH_4$ emissions have been adopted or are underway, for example in Colombia, the EU and North America (Environment and Climate Change Canada (2022), Ministerio de Minas y Energía (2022), U.S. Environmental Protection Agency (2023), EU Commission (2023)).

About a quarter of total global anthropogenic $CH_4$ emissions can be attributed to the oil and natural gas sector (Saunois et al. (2020)). The International Energy Agency (2023) suggests that deploying existing technological solutions could cut 75% of global oil and gas $CH_4$ emissions, at a cost less than 3% of the net income of the oil and gas industry in 2022. After the production segment, emission intensity is highest in the downstream distribution segment (Cooper et al. (2021), International Energy Agency (2023)). Regionally this sector can even present the largest share, especially in the EU where the vast majority of oil and gas used is imported (EU Commission (2023)).

Leak detection, quantification and verification surveys are key to mitigate $CH_4$ emissions from the oil and gas supply chain. $CH_4$ analyzers on various plattforms such as satellites, aircraft or vehicles have been utilized to detect and quantify fossil fuel related $CH_4$ emissions (Jacob et al. (2016), Cui et al. (2019), Maazallahi et al. (2020), Irakulis-Loitxate et al. (2021), Shen et al. (2021), Zavala-Araiza et al. (2021), Korbeń et al. (2022)). Both rapid detection and reliable quantification can help to improve and prioritize capital-intensive leak repair efforts (von Fischer et al. (2017)). This is especially important since leak distributions have been found to be skewed, a few large leaks are responsible for a major share of total emissions (Zavala-Araiza et al. (2015), Omara et al. (2016), Robertson et al. (2017), Weller et al. (2020), Maazallahi et al. (2020), Stavropoulou et al. (2023)). Beyond the direct mitigation opportunity, reliable quantification can help to evaluate the scale of fugitive emissions and possible emission reductions over time, related to the Global Methane Pledge. Vehicle-based mobile measurements deploying fast $CH_4$ gas analyzers have proven to be an efficient method for quickly and effectively surveying distribution networks across urban areas. There is currently no universally accepted measurement and data analysis methodology in place. Most widely used is the statistical methodology developed by von Fischer et al. (2017) and later refined by Weller et al. (2019). Based on controlled release experiment data (von Fischer et al. (2017)), they derived the following relation between the observed maximum excess $CH_4$ (in ppm) when crossing a plume and the release rate ($r_E$, in $\mathrm{Lmin^{-1}}$)

$$\ln([CH_4]_{\max}) = 0.817 \cdot \ln(r_E) - 0.988. \tag{1}$$

In the original multivariate regression model suggested by von Fischer et al. (2017) the release rate was used as the response variable and maximum enhancement, integrated spatial peak area and an index of the plume kurtosis were used as predictors.



Weller et al. (2019) argued that maximum $CH_4$ enhancement proved to be the best predictor of the release rate and that additional predictors did not meaningfully improve emission rate estimates.

This transfer equation has been applied in several measurement campaigns in North America and Europe (Phillips et al. (2013), von Fischer et al. (2017), Weller et al. (2018), Ars et al. (2020), Maazallahi et al. (2020), Defratyka et al. (2021), Fernandez et al. (2022), Wietzel and Schmidt (2023), Vogel et al. (2024)). During each survey, hundreds to thousands of leak indications were detected. The emission estimates were also used to rank leak indications into different categories (high ($>40$ $Lmin^{-1}$), medium (6-40 $Lmin^{-1}$) and small ($<6$ $Lmin^{-1}$) emissions), to assist with repair prioritisation decisions

(von Fischer et al. (2017), Maazallahi et al. (2020)). However, several problems have been identified. Leak indications cannot always be confirmed through re-measurement, and gas providers cannot always identify the source of the emissions (von Fischer et al. (2017), Weller et al. (2018), Maazallahi et al. (2020)). Furthermore, environmental factors such as wind, but also dispersion and turbulence within the emission plume lead to large uncertainties in measurements with corresponding over- and underestimations. Especially, within urban built environments complex wind flow, re-circulation and dispersion

patterns play an important role (Carpentieri and Robins (2015), Ražnjević (2023)). In addition, the approach developed in von Fischer et al. (2017) was derived for measurements using one specific device (Picarro G2301). By now multiple $CH_4$ analyzers are in use and it has been shown that $CH_4$ enhancements measured with different instruments can differ substantially (Maazallahi et al. (2023), Gillespie et al. (2023)). These discrepancies are expected, as different analyzers exhibit distinct instrument characteristics. Parameters such as cell volume, cell temperature, cell pressure, measurement frequency, and flow

rate vary among instruments, all of which influence the detected shape of the $CH_4$ peak (Takriti et al. (2021)). Thus the transfer function developed by Weller et al. (2019), where estimated emission rates purely depend on the measured peak maximum is not transferable to other analyzers. Maazallahi et al. (2020) noted that the integrated spatial peak area is much more consistent between different instruments than the peak maximum. Ars et al. (2020) found the maximum amplitude of $CH_4$ enhancement to be $50-80\%$ lower when sampled with a high inlet at 2.5 m compared to sampling close to the ground, whereas the integrated

spatial peak area was better comparable between the measurements at different heights.

This study evaluates results from different controlled $CH_4$ release experiments in four cities. Predictors for statistical emission rate estimation are evaluated. Specifically, we evaluate the consistency between measured peak heights and spatial peak areas across different instruments when transecting the same emission plume with the same air inlet. We then propose an instrument independent transfer equation that uses the integrated spatial peak area. We evaluate how successful this transfer

equation is in categorizing emissions into different categories, for single and multiple passes.

## 2  Methods

### 2.1  Controlled Release Experiments

Controlled release experiments (CRE) were conducted with a total of nine different analyzers in four different cities (London, Rotterdam, Toronto and Utrecht) situated in three different countries by different research groups (Table 1). The release lo-



cations are visualized in the SI, Figure S1 and an overview of the specific release rates per release location is given in Table A1.

**Table 1.** Controlled release experiments: Overview of cities, release rates, inlet height and location.

| CRE (City&Experiment) | GHG Analyzer | Release Rate Range $[\mathrm{Lmin}^{-1}]$ | Inlet Level [m] | Terrain |
|---|---|---|---|---|
| London, UK, Day-1 | G2301-m[a], uMEA[b] | 35,70 | 1.8 | Open field |
| London, UK, Day-2 | G2301-m[a], LI-7810[c] | 35,70 | 1.8 | Open field |
| London, UK, Day-3 | G2301-m[a] | 70 | 0.3, 1.8 | Open field |
| London, UK, II Day-1 | LI-7810[c] | 1 - 70.5 | 1.8 | Open field |
| London, UK, II Day-2 | LI-7810[c] | 0.2 - 1 | 1.8 | Open field |
| Rotterdam, NL | G2301[a], G4302[a], TILDAS[d], Mira Ultra[e], MGA10[f] | 0.15 - 120 | 1.7, 3 | Sub-urban |
| Toronto, CA Day-1 | G2401[a], UGGA[b] | 2.5 - 20 | 1.6, 2.5 | Industrial area |
| Toronto, CA Day-2 | G2401[a] | 0.1 - 5 | 2.5 | Parking lot |
| Utrecht, NL | G2301[a], G4302[a] | 2.2 - 15 | 0.5 | Urban |
| Utrecht, NL, II | G2301[a], Mira Ultra[e] | 0.15 - 100 | 0.5 | Urban |

[a]Picarro INC, Santa Clara, USA. [b]Los Gatos Research, San Jose, USA. [c]LI-COR Environmental, Lincoln, USA. [d]Aerodyne Research, Billerica, USA. [e]Aeris Technologies, Eden Landing Road Hayward, CA. [f]MIRO Analytical AG, Wallisellen, CH.

The controlled release experiment in Rotterdam was conducted on September 6, 2022. The location was selected to reflect common urban characteristics with houses, parked cars and overhanging trees (see SI, Figure S1a). Methane (purity > 99.9%) was released from two cylinders placed at a total of three locations along two connected streets at a wide range of flow rates

(0.15 - 120 $\mathrm{Lmin}^{-1}$). However, at two of the three locations the rotameter used to adjust the release rate was suspected to not work properly. Therefore only releases at location 1 (flowrate controlled electronically by an Alicat MCP-100SLPM, 5 - 120 $\mathrm{Lmin}^{-1}$) are considered. Atmospheric $CH_4$ mole fraction was measured while driving along the release locations, using five different instruments distributed over two vehicles. The first vehicle is Utrecht University's Air Quality car (UUAQ, Institute for Risk Assessment Sciences, Utrecht University), an Opel Astra with an air inlet on its roof (inlet height ca. 1.7 m, see SI,

Figure S2a). This car contained two cavity ring-down spectroscopy (CRDS) analyzers, model G2301 and G4302 (Picarro INC, USA) and a mid-infrared laser absorption spectroscopy analyzer MIRA Ultra (Aeris Technologies, CA). Two instruments, a MGA10 analyzer (MIRO Analytical AG, CH) and TILDAS Dual Laser Trace Gas Analyzer (Aerodyne Research, USA), were utilized in the measurement trailer of a truck operated by TNO. The inlet is on the side of the trailer around 2.5 m above ground level. In the morning, both vehicles drove separately. During the afternoon session, the G4302 and Mira Ultra analyzer were



transferred to the TNO semi-mobile truck to facilitate better comparison between the measuring devices and the UUAQ vehicle ceased its mobile measurements.

Two experiments in Utrecht were conducted at the Utrecht Science Park with multi-storey office and service buildings to the sides of the street (see SI, Figure S1b). On November 25, 2022, $CH_4$ was released simultaneously at two different locations (three different release rates spanning from 2.18 - 15 $Lmin^{-1}$). $CH_4$ mole fractions were measured by the G2301 and G4302 devices, the same devices used during the Rotterdam campaign, on board the UUAQ car. On June 11, 2024 $CH_4$ was released simultaneously at in total three different locations (0.15 - 100 $Lmin^{-1}$) and measured by the G2301 and Mira Ultra instrument installed in the IMAU van (Maazallahi et al. (2020)).

Nearby London (on an open airfield near Bedford), two measurement campaigns were carried out, the first on September 10-13, 2019 and the second on May 13 and 14, 2024. A G2301 analyzer was used on all days in the first campaign, on September 10 additionally an Ultraportable Methane:Ethane Analyser (uMEA, Los Gatos Research, San Jose, CA)) and on September 11 a LI-7810 $CH_4$/$CO_2$/$H_2O$ Trace Gas Analyzer (LI-COR Environmental, USA) were used. For the second campaign only data collected by the LI-7810 instrument were evaluated for this study. The driving pattern consisted of multiple parallel legs perpendicular to the estimated wind direction.

In Toronto, two CREs were carried out. In the Toronto industrial port lands neighbourhood, both a mobile bicycle-trailer-based laboratory equipped with a UGGA analyzer (inlet at 1.6 m above ground), and a vehicle based setup measuring with a G2401 analyzer (Picarro INC, USA, inlet at 2.5m above the ground) were deployed on October 20, 2021. The second experiment was carried out on October 24, 2021 on a parking lot, deploying the same vehicle-based set-up.

The average driving speed during plume transects was $3.8\pm1.0$ $ms^{-1}$, $5.6\pm0.9$ $ms^{-1}$, $5.9\pm0.9$ $ms^{-1}$, $3.0\pm0.5$ to $3.9\pm0.4$ $ms^{-1}$, $6.0\pm1.5$ to $7.3\pm1.4$ $ms^{-1}$ and $4.0\pm0.4$ to $7.5\pm0.9$ $ms^{-1}$ in Rotterdam, Utrecht, Utrecht II, London, London II and Toronto respectively. The median distance between the location where the plumes were detected and release location was 20 m, 20 m, 21 m, 20 to 25 m, 21 to 22 m and 17 to 24 m, which is typical for urban gas distribution networks (see SI, Sect. S6). More detailed descriptions of the measurement devices and experimental set-up can be found in the Supplementary Information, Maazallahi et al. (2020), Gillespie et al. (2023) and Vogel et al. (2024).

## 2.2 Data Treatment, Peak Identification, Determining Peak Maximum and Spatial Peak Area

The raw measurements were calibrated and corrected for inlet delay and a delay between different instruments (see SI, Sect. S3). A centred 5 minute moving window was applied to determine the atmospheric $CH_4$ background level at each point in time. The background level was defined as the $10^{th}$ percentile of the $CH_4$ mole fraction measurements, which was assessed to represent the background well (comparison is given in SI, Sect. S5). A peak was identified as a $CH_4$ enhancement reaching above 102% of background level. $CH_4$ enhancements in the calibrated $CH_4$ dataset were detected utilizing the python *scipy* function *find_peaks* (Virtanen et al. (2020)). Those individual peaks were then manually quality-checked for overlap of peaks, flawless function of all instruments deployed, validity of the transect and car speed. When multiple instruments recorded measurements in one vehicle the peak finding algorithm was applied to only one of the instruments. With a manual quality check it was ensured that the peak was valid for all instruments. For the London dataset peaks obtained at a distance further





than 75 m from the source were omitted to keep the distance within the same limits as for the other CREs. The maximum
$CH_4$ enhancement within the time interval of each peak was determined for each instrument. The peak finder algorithm was
applied to the G4302 device for the measurements on the UU car in Rotterdam and in Utrecht I, the G2301 instrument for
Utrecht II, the MGA10 device for the measurement on the TNO truck in Rotterdam, the uMEA and G2301-m device during
the CRE in London for Day 1 and Day 2 respectively. The peak area was integrated over space rather than time to take different
driving speeds into account. To convert the time series to the spatial coordinate, the $CH_4$ mole fraction enhancement at time
$t_{i+1}$ ($c(t_{i+1})$, in ppm, after subtraction of the $CH_4$ background level) was multiplied with the measurement time step of the
individual instrument ($t_{i+1} - t_i = \Delta t$) and the velocity of the vehicle averaged over the whole peak duration, yielding the
integrated spatial peak area in $ppm * m$.

$$[CH_4]_{area} = \sum_{i=0}^{n} \Delta t \cdot c(t_{i+1}) \cdot \bar{v}_{peak} \tag{2}$$

It is important to note that this spatial peak area does not correspond to the integration of $CH_4$ enhancement of the physical $CH_4$
plume in the environment across a 2D plane in space. Rather it represents a linear 1D fraction of the plume, that is described
by the driving track.

## 2.3 Emission Rate Estimation

An ordinary least-squares regression model (*scipy.stats.linregress* library, Virtanen et al. (2020)) was applied to the spatial peak
areas of the combined dataset. The natural logarithm (ln) of the known release rate was used as the independent variable and
the ln of integrated spatial peak area of $CH_4$ enhancements as the dependent variable.

$$\ln([CH_4]_{area}) = a_1 \ln(r_E) + a_0 \tag{3}$$

To assess the conformity of the data with assumptions underlying a linear regression (normality, linearity, independence
and homoscedasticity), several analyses were carried out (SI, Sect. S7). A similar fit was applied to the maximum excess $CH_4$
($\ln([CH_4]_{max}) = a_1^{max} \ln(r_E) + a_0^{max}$) for comparison to the equation from Weller et al. (2019).
To infer emission rate estimations based on measurements, the linear regression needs to be solved for the release rate.
Then the equation can be applied to measurements (the Weller eq. to the peak maximum measurements, the Area eq. to the
corresponding area measurements, for both cases background levels subtracted from $CH_4$ measurements). Sometimes the
algorithm produces estimates far outside the calibration range of the method, therefore a cap of $200 \, Lmin^{-1}$ was imposed on
emission rate estimations for the following evaluation.

## 2.4 Evaluating Quantification Performance

### 2.4.1 Categorization

To stay consistent with previous studies, release rates were classified into four different categories ($< 0.5 \, Lmin^{-1}$-Very low,
$0.5 - 6 \, Lmin^{-1}$-Low, $6 - 40 \, Lmin^{-1}$-Medium and $> 40 \, Lmin^{-1}$-High). For each leak indication ($i$) emission rates were



estimated utilizing both estimation methods, using the inverse of Equation 1 and Equation 3

$$r^i_{\text{E,Weller eq.}} = \exp\left(\frac{1}{a_1^W}\left(\ln([CH_4]_{\text{max}})^i - a_0^W\right)\right) = \exp\left(\frac{1}{0.817}\left(\ln([CH_4]_{\text{max}})^i + 0.988\right)\right) \tag{4}$$

$$r^i_{\text{E,Area eq.}} = \exp\left(\frac{1}{a_1^A}\left(\ln([CH_4]_{\text{area}})^i - a_0^A\right)\right) \tag{5}$$

The superscript W (Weller eq.) and A (Area eq.) differentiate the regression parameters. Subsequently, an estimated category was assigned to each peak given these inferred emission rates. This approach follows von Fischer et al. (2017), but is extended by a category for very small emissions as used in Vogel et al. (2024). For each group of peaks belonging to the same
category, the percentage of correctly classified peaks was calculated, along with the percentages of peaks that were erroneously categorized into other categories.

### 2.4.2 Percentage Difference in Emission Estimation as Function of Number of Transects

As will be shown below, variability in the plume shape causes large differences in observed peak maxima, spatial peak areas and thus derived emission rates for individual transects at the same actual emission rate. This variability can be reduced by
evaluating the average of several transects at the same emission rate. Following the analysis in Luetschwager et al. (2021) the effect of number of detections per $CH_4$ source on variability in estimated emission rate was explored.

Two approaches were followed to evaluate the emission quantification:

1. Comparison against the mean emission rate calculated from the leak indications (following Luetschwager et al. (2021)).

2. Comparison against the true release rate, which is known for our experiments.

To calculate the mean emission rates for the former comparison, the average natural logarithm of the integrated spatial peak area among all observed instances associated with each release rate $j$ was computed.

$$\text{Mean } \ln([CH_4]_{\text{area}})^j = \frac{1}{n}\sum_{i=1}^n \ln([CH_4]_{\text{area}})^i. \tag{6}$$

For each release rate, one mean emission rate $r^j_{\text{E,mean}}$ was obtained by applying the Area eq. to the calculated mean $\ln([CH_4]_{\text{area}})$. Subsequently, a Monte Carlo simulation was performed where we randomly selected between 2 and 10 emission peaks $i$ at
each release rate, and averaged the natural logarithm of the integrated spatial peak areas. Measurements obtained by different instruments during the same transects were treated as separate peaks. The theoretical number of possible subsets N from a certain size of set M with replacement is $\binom{M+N-1}{N} = \frac{(M+N-1)!}{N!\cdot(M-1)!}$. This gives for example between $465$ and $6\cdot10^7$ combinations for $N$ between 2-10 and $M = 30$. This procedure was repeated 2000 times for each release rate and number of detections. That means for $N = 3$, three peaks were randomly sampled 2000 times from a given release rate, yielding 2000 emission rate
estimations $r^{k,j,N}_{\text{E,sim}}$ ($k \in [1, 2000]$) for each release rate. For each of those 2000 emission rate estimations $r^{k,j,N}_{\text{E,sim}}$ the percentage





difference to the mean emission rate $r^j_{\mathrm{E,mean}}$ and the known release rate $r^j_{\mathrm{E,true}}$ were calculated:

$$\text{Percentage Deviation } \Delta\%^{k,j,N}_{\mathrm{mean}} = \frac{r^{k,j,N}_{\mathrm{E,sim}} - r^j_{\mathrm{E,mean}}}{r^j_{\mathrm{E,mean}}} \cdot 100\% \tag{7}$$

$$\text{Percentage Deviation } \Delta\%^{k,j,N}_{\mathrm{true}} = \frac{r^{k,j,N}_{\mathrm{E,sim}} - r^j_{\mathrm{E,true}}}{r^j_{\mathrm{E,true}}} \cdot 100\%. \tag{8}$$

Then, an average percentage deviation for each release rate $j$ and number of transects $N$ was determined for both cases

Mean Percentage Deviation $\overline{\Delta\%}^{j,N} = \frac{1}{2000} \sum_{k=1}^{2000} \Delta\%^{k,j,N}.$ (9)

Finally, an average over the different release rates $J$ for each number of transects $N \in [2, 10]$ was calculated for both cases

$$\text{Overall Mean Per N} = \frac{1}{J} \sum_{j=1}^{J} \overline{\Delta\%}^{j,N} \tag{10}$$

The analysis was done for each unique release rate and location pair. Experiments with fewer than 10 transects were filtered out, as the Monte Carlo analysis samples up to 10 measurements (N ranging from 2 to 10) from the available transects. This

leaves 35 of the original 55 release rates. Lastly, another categorization analysis was conducted with the goal to investigate the classification performance when incorporating multiple transects. For each of the 2000 mean emission rates per release rate and number of transects, a category was assigned and evaluated.



# 3 Results and Discussion

## 3.1 Instrument Performance: Peak Maximum and Spatial Peak Area



(a) Peak Maximum
(b) Spatial Peak Area

**Figure 1.** Comparison of peak maximum (a) and spatial peak area (b) from different instruments deployed in Rotterdam and Utrecht (subscript '_U'), shown are the data points and a linear regression fit with intercept 0 for each instrument. The results from the G2301, Mira Ultra, MGA10 and TILDAS devices are plotted on the y-axis and the results from the G4302 instrument on the x-axis. The black dotted line represents the 1:1 line. (For the G2301 analyzer, peaks exceeding a maximum of 20 ppm are marked with an 'x' and excluded from the fitting process.)

Figure 1a shows that the comparison of measured peak maxima among different instruments reveals very strong systematic discrepancies. Compared to the G4302, all other instruments (Mira Ultra, G2301, MGA10, and TILDAS) strongly underestimate the peak maximum. Specifically, the maxima recorded by the TILDAS are only half compared to those of the G4302, while




the Mira Ultra, G2301, and MGA10 show peak maxima that typically are only between $10\%$ to $30\%$ of the G4302 readings.
Figure 1b shows that the evaluated spatial peak areas are much more consistent between instruments with slopes between 0.63

and 1.01. The coefficient of determination $R^2$ is notably higher for the area fit when compared to the maximum fit for most
instruments. $R^2$ for G2301_U, Mira Ultra, and TILDAS all exceed 0.96. The MGA10 device stands out as an exception, as it
demonstrates poor $R^2$ values for both maximum and area fits. Both, measured maximum and area values are more scattered
and exhibit strong deviations from the other instruments. Interestingly, the G2301 analyzer aligns better with the G4302 for
data collected in Utrecht than in Rotterdam (indicated by a higher slope). This reflects the better alignment of instrument mea-

surements for smaller peaks; in Rotterdam very high release rates were included. A similar pattern arises for the evaluation
of the CRE in London (Figure B1). Both the uMEA and LI-7810 instrument measurements align more closely with G2301
measurements when assessing spatial peak area rather than peak maximum.

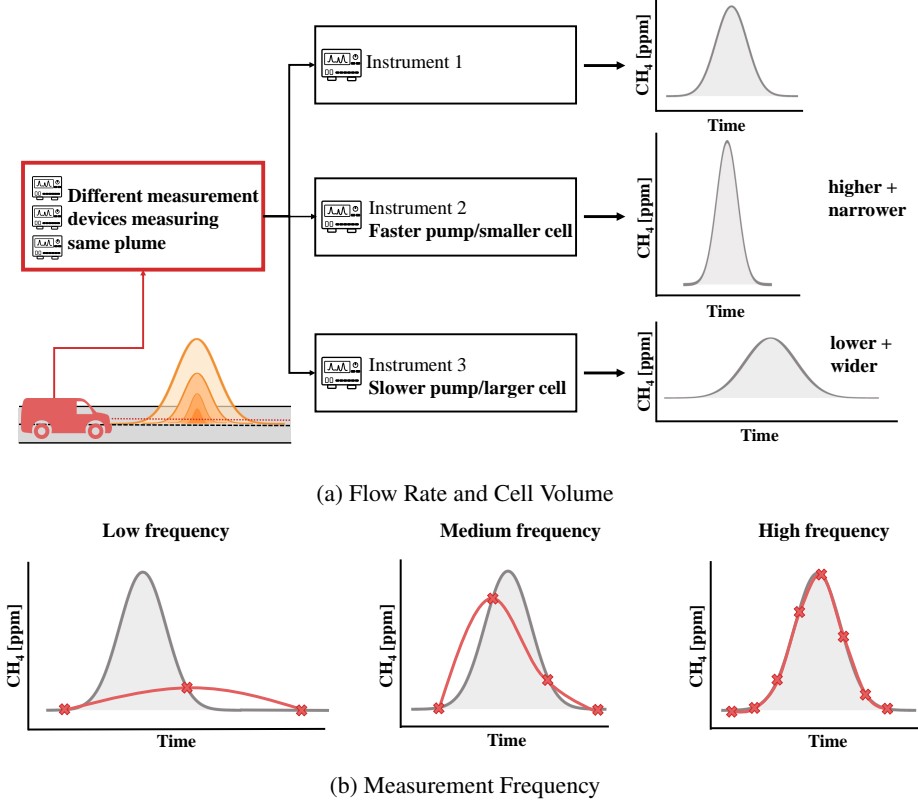

(a) Flow Rate and Cell Volume

(b) Measurement Frequency

**Figure 2.** Influence of the instrument characteristics flow rate, cell volume (in dependence of cell temperature and pressure) and measurement frequency on the peak shape detected.

Our analysis demonstrates that different $CH_4$ analyzers show much better consistency in the integrated $CH_4$ spatial peak
area compared to the maximum $CH_4$ enhancement. This finding is supported by conceptual considerations: Measured mole

fractions are influenced by instrument characteristics, mainly flow rate, cell volume and measurement frequency (Figure 2).





A higher flow rate or smaller cell volume will generally make measured mole fraction profiles sharper and higher. Lower flow rates and larger cell volumes on the other side lead to smoothing effects, rendering the $CH_4$ peak smaller and wider (Takriti et al. (2021), Maazallahi et al. (2023)). While the peak maximum is strongly affected by those characteristics, the integrated area is not. Each molecule that enters an instrument is measured, either as part of a wide and low or a narrow and high peak. Maazallahi et al. (2023) and Gillespie et al. (2023) made similar observations when comparing different analyzers. Beyond differences in instrument performance, Ars et al. (2020) noted the area to be more robust against differences in inlet height, further supporting the argument for using the integrated spatial peak area as a more reliable metric for quantifying $CH_4$ enhancements. In the future, even more instruments are expected to be deployed across local natural gas distribution networks. It is therefore important to move beyond the maximum enhancement as emission estimation metric and use the spatial peak area instead.

### 3.2 Converting Spatial Peak Areas to Emission Rates

Figure 3b shows the recorded spatial peak areas as a function of the known release rates (double logarithmic axes) from all controlled release experiments. As anticipated, higher $CH_4$ release rates in general correspond to higher observed spatial peak area measurements. A linear fit can be applied to these data and the fit equation is proposed as empirical model relating integrated spatial peak area of $CH_4$ enhancements (in $ppm*m$, background levels subtracted from $CH_4$ measurements) to emission rates (in $Lmin^{-1}$):

$$\ln([CH_4]_{area}) = 0.774 \cdot \ln(r_E) + 1.84. \tag{11}$$

In practice, emission rate estimations will be derived from area measurements. To achieve this, Equation 11 can be solved for $r_E$.

$$r_E = \exp\left(1.292 \cdot \ln([CH_4]_{area}) - 2.377\right) \tag{12}$$

The natural logarithm of the spatial peak area associated with a leak indication can then be inserted in the equation. If multiple transects were taken, the average logarithm of the spatial peak area values $\overline{\ln([CH_4]_{area})}$ corresponding to the same leak indication should be used.

A similar linear regression fit was applied to peak maxima data: $\ln([CH_4]_{max}) = 0.854 \cdot \ln(r_E) - 1.25.$ (Figure 3a). The inferred regression equation for the maximum shows good agreement with the model proposed by Weller et al. (2019).

### 3.3 Using Survey Data for Repair Prioritisation

A wide range of peak maxima and areas is observed for each release rate. This illustrates the shortcomings of the quantification of emissions for individual leak indications using this simple statistical method. This spread reflects the nature of turbulent dispersion of plumes. Environmental factors, in particular built environment and meteorology, e.g. convection, complex wind patterns within urban areas (back circulation, wind channelling, blockages), turbulence and diffusion, play a major role in determining the location, shape and $CH_4$ mole fraction of the meandering $CH_4$ plume (Carpentieri and Robins (2015), Cassiani




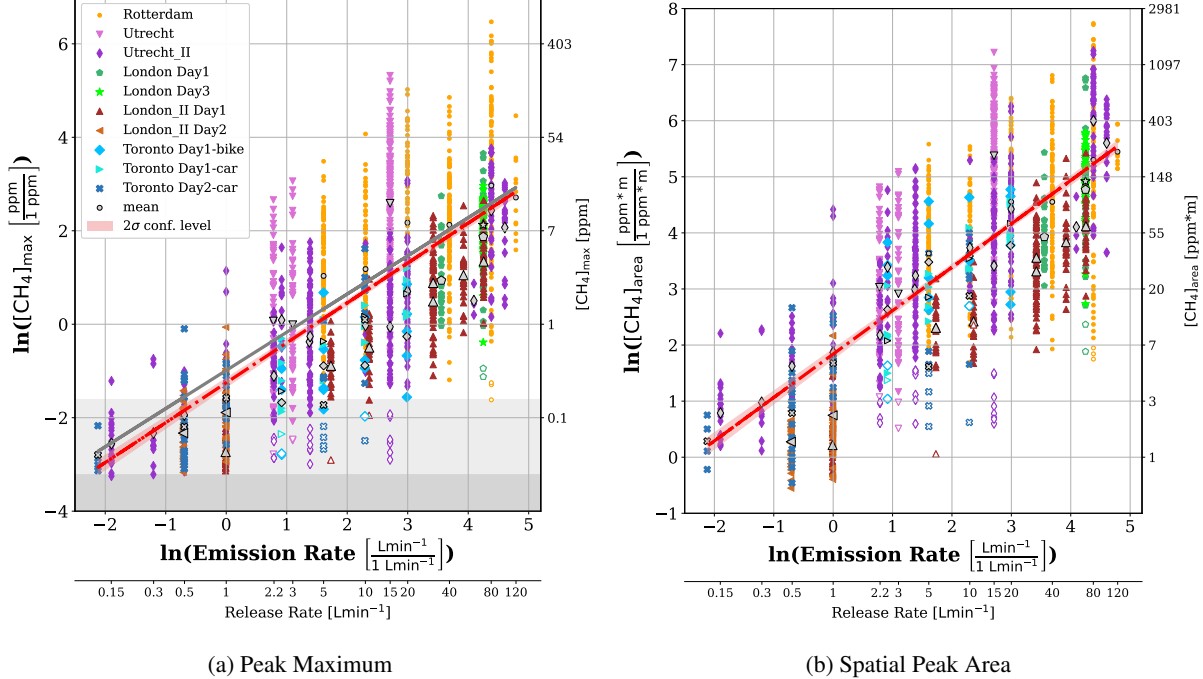

(a) Peak Maximum                                    (b) Spatial Peak Area

**Figure 3.** Correlation of the natural logarithm of the peak maximum enhancement (a) and spatial peak area (b) with the natural logarithm of the release rates for all controlled release experiments reported in this manuscript (except London Day 2). Different cities are indicated by different colours (Rotterdam-orange, Utrecht I-pink, Utrecht II-dark purple, Toronto-blue, London I-green and London II-brown). Black markers indicate mean values per release rate and city, unfilled markers indicate potential outliers. The second x-axis indicates release rates deployed, the red lines are linear regressions to all data. The Weller equation is displayed in gray as a comparison (a) and the light (dark) gray area indicates peaks below 110% (102%) of background level.

et al. (2020), Ražnjević et al. (2022)). Still, a greater $CH_4$ release increases the likelihood of detecting higher $CH_4$ mole fractions. This is the relation that is captured in the proposed transfer equation. Yet, changing wind conditions or turbulence can still cause the main plume to become highly diluted or be transported away from the street, leading to smaller peak maxima

and spatial peak areas. At the extremes, high $CH_4$ mole fractions are measured in one transect and no peak at all in another one (Luetschwager et al. (2021)).

Additionally, some assumptions underlying the application of a linear regression are partially violated (homoscedasticity, normal distribution of errors). However, the premises were deemed sufficiently met to justify the use of linear regression, allowing for a straightforward and practical statistical inference model. However, as all experiments included in this study were

executed during daytimes (measurements were conducted between 09:00-18:00 local time) the method might not be suitable for the evaluation of nighttime measurements. During the night, the atmospheric boundary layer becomes more stable, which suppresses updrafts and leads to an accumulation of $CH_4$ emissions at the surface. This causes higher measured mole fractions,





increasing both the peak maximum and spatial peak area measurements; hence most likely considerable overestimation of the emission rate.

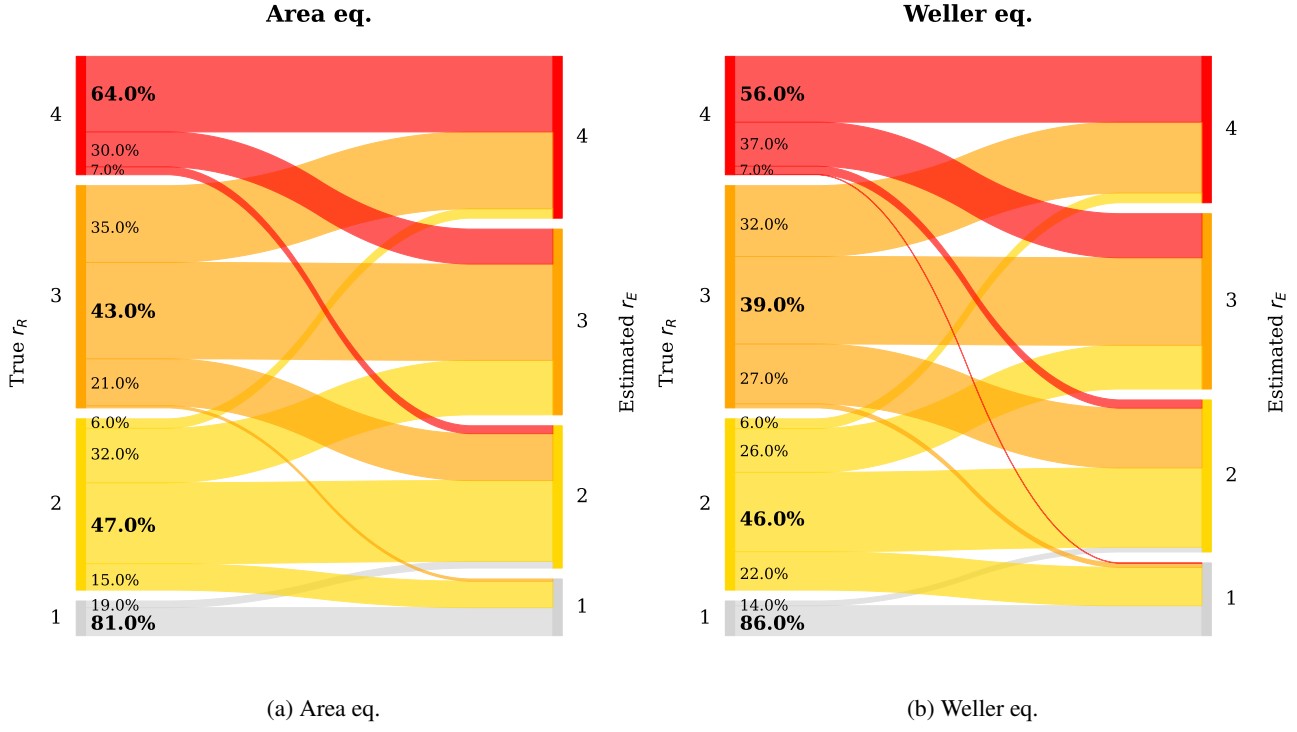

(a) Area eq.                                            (b) Weller eq.

**Figure 4.** Categorization performance of the Area and Weller method (including data from all 6 CRE). The left y axis represents the true emission rate $r_E$, where the width of the bars indicate the amount of plumes belonging to each emission category (categories: 1-Very Low, 2-Low, 3-Medium and 4-High). The right y axis represents the categories estimated by the statistical model and the connecting lines visualize the amount of plumes from each category pool which the algorithm classifies into another (or the same) category.

Deviation of emission rate estimation from true emission rate can be very high for an individual gas leak, especially for single transects. For our set of experiments under- and overestimation range from -100% to +2700%. To facilitate a pragmatic repair priorization, categorizing leaks can be an important tool. We quantified the emission rate of all leak indications and subsequently assigned them to one of the four categories described above. Figure 4 illustrates the categorization performance based on quantification using the spatial peak area (a) and peak maximum (b). It displays the percentage of peaks correctly

classified (bold number) and also shows in which categories the rest is falsely overestimated or underestimated. A suitable benchmark for categorization is a 25% accuracy rate, reflecting the expected success when peaks are randomly assigned to one of the four categories.

    The majority of very low and high peak indications are correctly attributed. Peaks in the medium category are less well classified, with a large portion being overestimated. Overall, the proportion of peaks that are misclassified by more than one

category is very low. Around 7% (22%) of category 4 (3) peaks are underestimated into category 1 or 2, when applying the area





eq. This means that the vast majority (78% to 93%) of higher emission peaks from categories 3 and 4 are correctly identified as high, effectively differentiating them from low emission peaks. At the same time, 38% (0%) of category 2 (1) emission rates are falsely identified as high emission rates. The same analysis using the Weller eq. yields comparable results, with a slightly lower categorization performance except for the lowest emission category.

It is important to note that these categorization performance rates vary across our set of release locations and are therefore not directly transferable to any leak populations in urban areas. The quantification and categorization performance differs for different locations, for example the categorization precision for the London data is 62% for category 4, but 88% for category 3, much higher than for the average (SI, Figure S22). Given the local circumstances regarding built environment and meteorological conditions peak measurements can be systematically lower or higher compared to the mean of our dataset.

However, the numbers can still be considered an indication of typical categorization performance.

In von Fischer et al. (2017) a categorization success rate of over 80% for category 2 and 3 was reported, which is higher than for our dataset. For the highest category their performance was lower (38%) than ours (64%). The differences might be due to the specific local conditions where their controlled release took place. In practice, $> 80\%$ of urban leaks have a low emission rate (von Fischer et al. (2017), Maazallahi et al. (2020), category 1 was not assessed in those publications).

Overall, our results show that the categorization approach for leak repair prioritization would perform far better than making repair decision by chance. Therefore, this approach offers a potential to reduce emissions since larger leaks can be targeted first. The Area eq. does not consistently outperform the Weller model in terms of categorization performance, but it yields more consistent results across different instruments, which would facilitate comparison of measurement surveys performed with different instruments.

## 3.4    Benefit of Multiple Transects

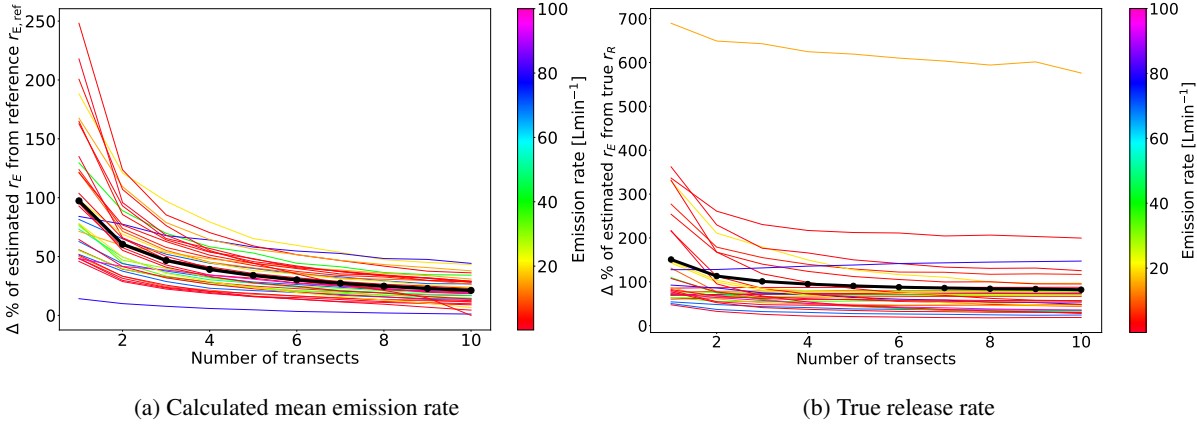

(a) Calculated mean emission rate                    (b) True release rate

**Figure 5.** Mean absolute percentage deviation of emission estimation from the (a) calculated mean emission rate and (b) true release rate as a function of the number of transects. Each line represents one release rate at one location, and was created by a Monte-Carlo analysis with 2000 repetitions. The total mean over all release rates considered in this analysis is shown in black.





The discussed high uncertainty of individual estimations of the emission rate suggests that the performance may be improved by carrying out several transects. Including more measurements will reduce the random error associated with measurements and increase the confidence to capture the mean of this particular distribution (Cochran (1977)). This effect has been demonstrated in Luetschwager et al. (2021) and is illustrated in Figure 5a. The percentage deviation from the mean at a certain release location and release rate decreases drastically with an increasing number of transects.


The observation that many of the mean values for individual release rates are still considerably different from the value expected from our conversion equation (Figure 3) suggests that in addition to random error, there is also a systematic error. This is likely due to two factors; the offset of the sample mean from the population mean (determined by longer time scale weather phenomena) and the offset of the population mean from our linear regression (determined by (built) environment).


Therefore, it follows that the error with respect to the true release rate will not necessarily decrease with more measurements for mean values that are far apart from the regression line. This is the case for the data from London II experiment, which exhibit a large negative offset as well as for the Utrecht 15 $\mathrm{Lmin}^{-1}$ release, which exhibits a very high positive offset (see SI, Sect. S9 for a more detailed discussion). Still, if the offset is not too large, the error in emission estimation decreases with more transects (Figure 5b). Note that here the absolute percentage error is displayed, no difference is made between a negative or

positive deviation.

The largest decrease in emission estimation error can be achieved from 1 to 3 transects, reducing the estimation error by one third. A further decrease can be reached until around 5-7 transects are included, after which the gained variation reduction levels off, including more transects then reduces the uncertainty by less than 3%.

Luetschwager et al. (2021) carried out a similar analysis on multiple passages of real-world leak locations, comparing

individual emission rate estimate to the emission rate estimate based on the distribution mean. They reported the sharpest decline in variation with increased detections from 2 to 4, aligning with our analysis. They also noted that beyond 5-6 transects, the reduction in variability diminishes. Additionally, Luetschwager et al. (2021) reported errors relative to the mean of 50% deviation based on 2 transects. This compares well with the variability in our corresponding analysis, which amounts to an average of 62% for N=2 comparing to the mean. Nevertheless, in comparison to true release rates from real measurements the

overall deviations are higher than in the aforementioned studies. We find the mean percentage deviation from the true release rate to be 115% for N=2. Further, the deviations are skewed with higher overestimations than underestimations, a result of the right-skewed distribution of measured spatial peak area values.

Ražnjević (2023) stated that at least 10 transects are necessary to estimate the source strength within 40% of the true emission, simulating a mobile measurement set-up in an Large Eddy Simulation model. We found for half of the cases an

estimation error within 58% when including 10 transects, for the other half errors ranged between 65-200% with one outlier at over 500%. Thus, our findings are in closer agreement with Ražnjević (2023) than with the smaller estimation errors reported in Luetschwager et al. (2021). The generally higher estimation errors demonstrate the complexity of real world environmental influences.

The persistent estimation error, even with the inclusion of more transects, likely reflects the influence of the built-environment

and meteorological conditions. If there is a systematic bias for that specific location or time frame of measurements, combining




several transects will still include that high or low bias. Additional measurements can only balance out random errors or fluctuations. Taking measurements under different meteorological conditions could potentially improve estimations, enabling sampling of the full distribution of possible realizations of the emission plume.

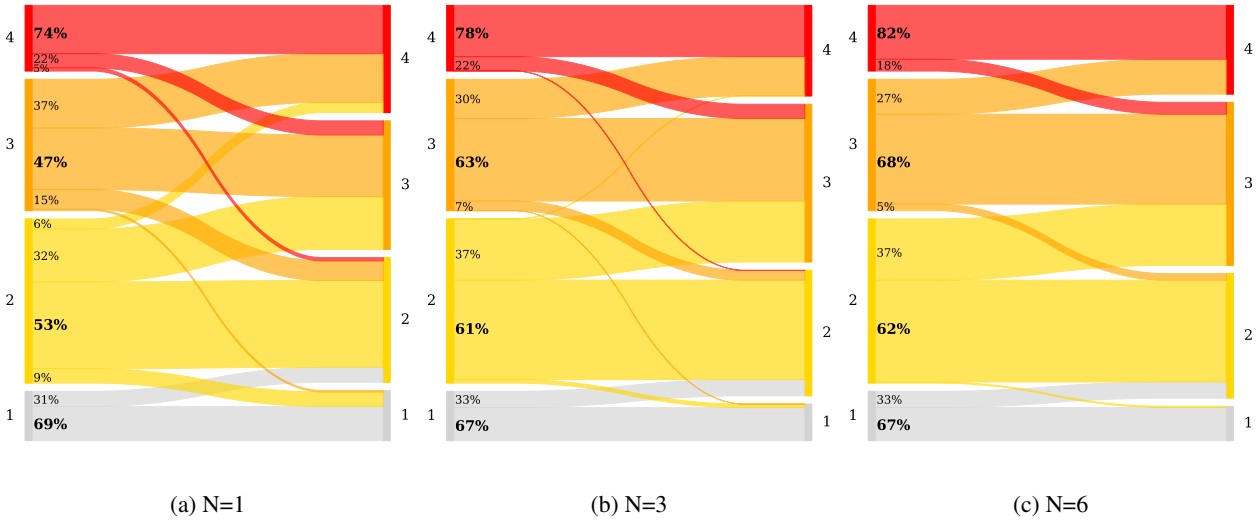

(a) N=1         (b) N=3         (c) N=6

**Figure 6.** Categorization performance of the area equation with (a) N=1, (b) N=3 and (c) N=6 transects for distributions with a small to medium offset (25 of the 35 release rates). The left axis represents the true $r_E$, where the width of the bars indicate the amount of plumes belonging to each emission category (categories: 1-Very Low, 2-Low, 3-Medium and 4-High). The right axis represent the categories estimated by the statistical model and the connecting lines visualize the amount of plumes from each category pool which the algorithm classifies into another (or the same) category. The underlying dataset is based on the Monte-Carlo simulation resulting in 2000 estimates per release rate.

Figure 6 illustrates that the accuracy of categorization on average improves when taking into account three (Figure 6b) or six (Figure 6c) transects. The London II and Utrecht 15 $\mathrm{Lmin}^{-1}$ releases are not included in this analysis because they include strong systematic biases.

For emission rate estimations based on a single detection the categorization performance ranges between 47% and 74%, averaging 58%. When three transects are used, the categorization performance improves to an average of 65%. Including six transects further enhances performance to an average of 69%, ranging from 62% to 82%. More importantly, the number of strong over or underestimations decreases. For example, whereas for only one transect 5% of category 4 peaks and 15% of category 3 peaks are categorized as category 2, this reduces to 0% and 5% respectively for 6 transects. Also the false overestimation of 6% of category 2 peaks into category 4 vanishes when increasing sampling effort to 6 measurements.

### 3.5 Suggested Method for Routine Implementation of Leak Surveys

We suggest the following approach for routine mobile leak detection surveys in urban areas:



1. Survey: Perform surveys with an instrument that falls under the '1-1-10 principle' - at least 1 Hz measurement frequency, 1 s residence time and 10 ppb precision. Record GPS data together with the mole fraction measurements, ideally with the same measurement frequency as the $CH_4$ analyzer. Ensure that the GPS and $CH_4$ recordings are time-synchronized. Measure and record the inlet delay time for the $CH_4$ measurements at the beginning of the survey. During the survey, if an enhancement above the $CH_4$ background exceeds 0.2 ppm, attempt to conduct six transects.

2. Evaluation: After the survey campaign, determine $CH_4$ background levels and subtract it from the $CH_4$ measurements ($CH_4$ background at a certain time = $10^{th}$ percentile of $CH_4$ mole fractions within a $\pm 2.5$ min time window). Use a peak detection algorithm to identify peaks in the time series, including start and end time of each peak. If available, use additional information such as ethane measurements, methane : carbon dioxide ratio or $CH_4$ isotopic composition to differentiate between biogenic, combustion and fossil sources. Use GPS data to infer the driving speed to convert the time series to space coordinates and calculate spatial peak areas (using the $CH_4$ elevation above background levels). In case several transects were done for the same leak indication, take the average of the $\ln([CH_4]_{area})$ of the individual peaks detected during the different transects. Calculate the emission rate by inserting the $\ln([CH_4]_{area})$ value into the inverse Area eq.

3. Categorization: Rank the leak into one of the four emission categories ($< 0.5$ $Lmin^{-1}$-Very low, $0.5 - 6$ $Lmin^{-1}$-Low, $6 - 40$ $Lmin^{-1}$-Medium and $> 40$ $Lmin^{-1}$-High) and use this categorization in addition to safety concerns to prioritize leak repair efforts.

4. Report: Report the number of leaks per category and number of leaks per pipeline kilometre. Report the leak size distribution and total emissions for the surveyed area. Ideally, provide these numbers separately for different pipeline material (e.g. cast iron, steel, polyethylene and P.V.C.).

## 4 Conclusions

Our results clearly highlight systematic differences in maximum $CH_4$ enhancement recorded by different commercial $CH_4$ gas analyzers. Maximum enhancements differ by up to a factor ten between instruments. When the spatial peak areas are evaluated, differences between instruments reduce to less than a factor two, confirming our hypothesis that the spatial peak area is a more suitable quantity than the recorded peak maximum. We suggest moving beyond the peak maximum as emission estimation metric to avoid biased emission predictions.

We propose an empirical equation for the emission rate (in $Lmin^{-1}$) based on integrated spatial peak area of $CH_4$ enhancements (in $ppm * m$, background levels subtracted from $CH_4$ measurements):

$$r_E = \exp\left(1.292 \cdot \overline{\ln([CH_4]_{area})} - 2.377\right) \tag{13}$$

Here, $\overline{\ln([CH_4]_{area})}$ is the mean taken over several transects associated with the same leak indication. This formula is very simple in deployment, it does not require any additional information than the actual $CH_4$ mole fraction and GPS measurements.





But, it therefore ignores other important local influences. The most important parameters are likely the built environment, wind speed and direction, atmospheric stability and turbulent processes. It is expected that incorporating parameters reflecting those factors could improve emission rate predictions. However, especially when it comes to incorporating turbulent processes, the associated measurement effort would become extensive. Therefore, the strength of our simple statistical model is the simplicity

of its implementation, still providing reasonable emission estimates. Higher sampling effort can improve quantification accuracy significantly. Including three transects improved correct categorization into four emission categories from 47%-74% to 61%-78% and to 62%-82% when incorporating six transects. Thus, applying this method to identify, quantify and prioritize leak repairs in urban natural gas distribution systems can support greenhouse gas emission mitigation during the transition to a fossil-fuel-free energy system.



## 385 Appendix A: Overview Settings of the Controlled Release Experiments

Table A1: Controlled release experiments: Overview of release rates, duration of releases per location, number of valid transects and number of valid peaks. In cases where several instruments were mounted on the same vehicle, the number of peaks is higher than the number of transects. The time is given as local time (difference to UTC: London UTC+01:00, London II UTC+01:00, Rotterdam UTC+02:00, Toronto UTC+04:00, Utrecht UTC+01:00, Utrecht II UTC+02:00, ). The number of transects/peaks with bike and car platforms are given separately for Toronto Day1 (bike + car).

| City | Location | Release Rate $[\mathrm{Lmin}^{-1}]$ | Duration (local time) | Valid Transects | Valid Peaks |
|---|---|---|---|---|---|
| London Day1 (September 10, 2019) | | | | | |
| | 1 | 70 | 11:01 - 15:42 | 36 | 72 |
| | 1 | 35 | 16:01 - 17:20 | 13 | 26 |
| London Day2 (September 11, 2019) | | | | | |
| | 1 | 35 | 12:16 - 13:05 | 15 | 30 |
| | 1 | 70 | 13:22 - 14:11 | 10 | 20 |
| | 1 | 70 | 16:28 - 17:20 | 22 | 44 |
| London Day3 (September 13, 2019) | | | | | |
| | 1 | 70 | 10:25 - 11:53 | 42 | 42 |
| London II Day1 (May 13, 2024) | | | | | |
| | 1 | 70.5 | 14:18 - 14:44 | 31 | 31 |
| | 1 | 50.5 | 15:03 - 15:31 | 29 | 29 |
| | 1 | 30.6 | 15:38 - 15:58 | 29 | 29 |
| | 1 | 10.6 | 16:14 - 16:40 | 24 | 24 |
| | 1 | 5.6 | 16:44 - 17:06 | 26 | 26 |
| | 1 | 1 | 17:14 - 17:36 | 22 | 22 |
| | 1 | 30.6 | 17:47 - 18:15 | 49 | 49 |
| London II Day2 (May 14, 2024) | | | | | |
| | 1 | 1 | 10:08 - 10:47 | 40 | 40 |
| | 1 | 0.5 | 11:02 - 11:32 | 34 | 34 |
| | 1 | 0.2 | 11:37 - 12:23 | 1 | 1 |



**Table A1 – continued from previous page**

| City | Location | Release Rate $[\mathrm{Lmin}^{-1}]$ | Duration (local time) | Valid Transects | Valid Peaks |
|---|---|---|---|---|---|
| Rotterdam (September 6, 2022) | | | | | |
| | 1 | 5 | 9:05 - 10:15 | 49 | 138 |
| | 1 | 10 | 10:15 - 10:58 | 35 | 97 |
| | 1 | 20 | 10:58 - 11:23 | 21 | 61 |
| | 1 | 40 | 11:23 - 11:54 | 34 | 97 |
| | 1 | 80 | 11:54 - 12:44 | 44 | 124 |
| | 1 | 20 | 13:05 - 13:34 | 6 | 24 |
| | 1 | 120 | 13:34 - 13:48 | 3 | 12 |
| | 1 | 40 | 13:48 - 14:26 | 6 | 24 |
| | 2 | 1 | 10:28 - 11:01 | 21 | 62 |
| | 2 | 0.15 | 11:01 - 11:39 | 20 | 59 |
| | 2 | 0.515 | 11:39 - 12:12 | 23 | 66 |
| | 2 | 0.31 | 12:12 - 13:16 | 22 | 66 |
| | 3 | 3.33 | 13:16 - 14:26 | 9 | 36 |
| Toronto Day1 (October 20, 2021) | | | | | |
| | 1 | 9.9 | 16:11 - 16:18 | 4+4 | 4+4 |
| | 1 | 5 | 16:19 - 16:27 | 5+4 | 5+4 |
| | 1 | 2.5 | 16:30 - 16:40 | 5+7 | 5+7 |
| | 1 | 19.8 | 16:40 - 16:49 | 5+5 | 5+5 |
| Toronto Day2 (October 24, 2021) | | | | | |
| | 2 | 9.9 | 9:48 - 9:58 | 7 | 7 |
| | 2 | 5 | 10:03 - 10:11 | 7 | 7 |
| | 2 | 1 | 10:16 - 10:24 | 11 | 11 |
| | 2 | 0.12 | 10:28 - 10:37 | 4 | 4 |
| | 2 | 0.5 | 10:41 - 10:59 | 19 | 19 |
| Utrecht (November 25, 2022) | | | | | |
| | 1 | 3 | 13:06 - 13:46 | 24 | 48 |
| | 1 | 2.18 | 14:22 - 15:17 | 28 | 56 |



**Table A1 – continued from previous page**

| City | Location | Release Rate $[\mathrm{Lmin}^{-1}]$ | Duration (local time) | Valid Transects | Valid Peaks |
|------|----------|------------------------------------|----------------------|-----------------|-------------|
| | 2 | 15 | 13:06 - 13:46 | 61 | 122 |
| | 2 | 15 | 14:22 - 15:17 | | |
| Utrecht II (June 11, 2024) | | | | | |
| | 1 | 4 | 10:48 - 11:22 | 10 | 10 |
| | 1 | 4 | 11:38 - 11:48 | 5 | 5 |
| | 1 | 10 | 11:48 - 12:18 | 6 | 6 |
| | 1 | 80 | 12:18 - 12:32 | 6 | 11 |
| | 1 | 20 | 12:32 - 13:01 | 23 | 46 |
| | 1 | 100 | 14:05 - 14:21 | 10 | 20 |
| | 1 | 15 | 14:28 - 15:23 | 32 | 64 |
| | 1 | 4 | 16:19 - 17:07 | 18 | 35 |
| | 1 | 0.15 | 17:07 - 17:44 | 2 | 2 |
| | 1 | 1 | 17:44 - 17:56 | 1 | 1 |
| | 2 | 2.5 | 10:50 - 11:31 | 16 | 16 |
| | 2 | 4 | 11:31 - 12:09 | 12 | 12 |
| | 2 | 0.5 | 12:09 - 12:37 | 13 | 20 |
| | 2 | 0.15 | 12:37 - 13:01 | 10 | 20 |
| | 2 | 0.3 | 14:02 - 14:40 | 5 | 10 |
| | 2 | 2.2 | 14:40 - 15:13 | 18 | 36 |
| | 2 | 1 | 15:13 - 15:52 | 19 | 38 |
| | 2 | 4 | 17:00 - 17:30 | 17 | 22 |
| | 2 | 60 | 17:30 - 17:38 | 3 | 3 |
| | 2 | 20 | 17:38 - 18:06 | 11 | 11 |
| | 2 | 80 | 18:06 - 18:25 | 16 | 32 |





## Appendix B: Instrument Performance: Peak Maximum and Spatial Peak Area

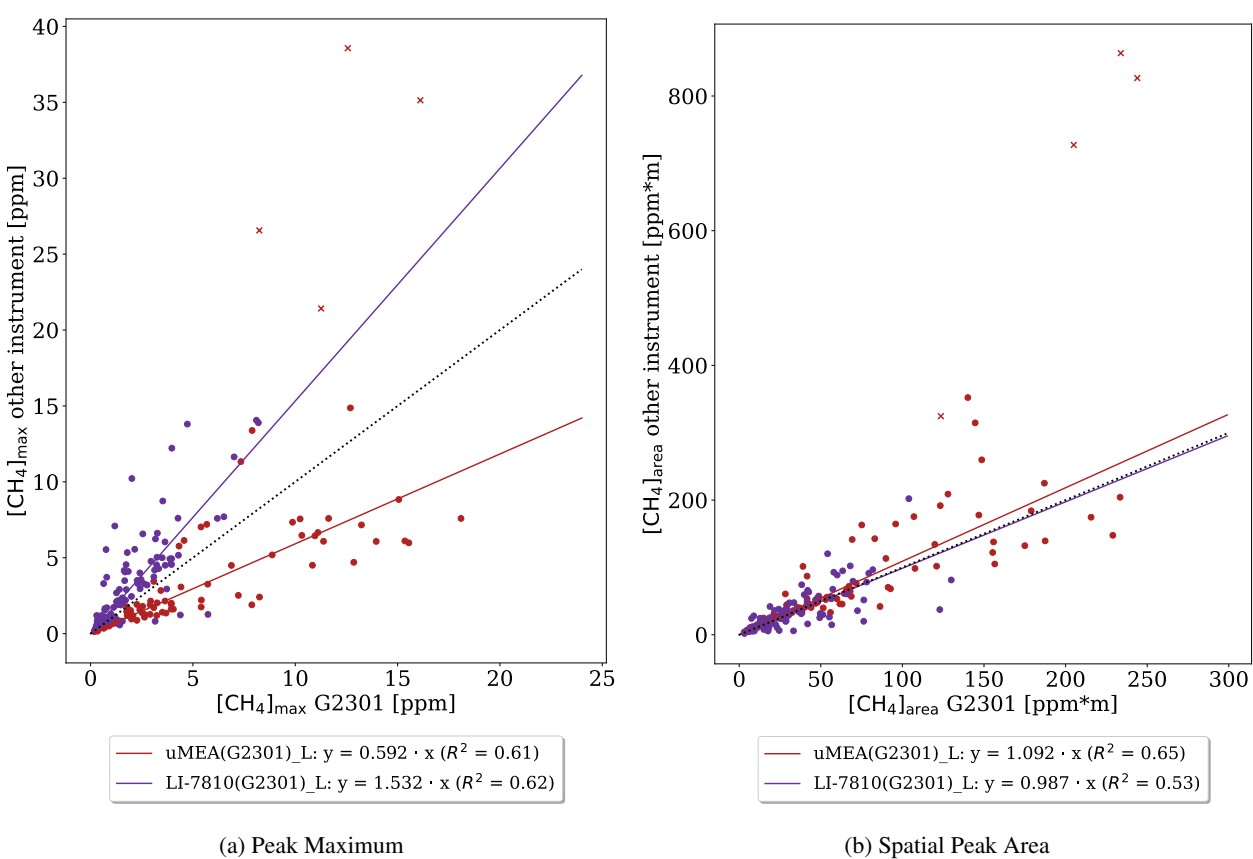

(a) Peak Maximum          (b) Spatial Peak Area

**Figure B1.** Comparison of peak maximum (a) and spatial peak area (b) from different instruments in London. Regression fits with intercept 0 are applied to the data for each instrument. The results from the uMEA and LI-7810 analyzers are plotted on the y-axis and the results from the G2301-m instrument on the x-axis. The black dotted line represents the 1:1 line. (Peaks exceeding a maximum of 20 ppm are marked with an 'x' and were excluded from the fitting process.)



*Code and data availability.* The python code and a sub-sample of the data used to produce the results in this article are available on GitHub: https://github.com/judith-tettenborn/CRE_CH4Quantification.git

*Author contributions.* Contributed to conception and design: T.R., D.Z.-A., H.M.

Contributed to acquisition of data: J.T., D.S, H.M, C. vd V., A.H., I.V., P. vd B., F.V., L.G., S.A., J.F., D.L., R.F., T.R.

Contributed to analysis and interpretation of data: J.T., D.Z.-A., D.S., H.M., A.H., J.F., T.R.

Drafted and/or revised the article: J.T., D.Z.-A., H.M, A.H., I.V., F.V., L.G., S.A., J.F., D.L., R.F., T.R.

Approved the submitted version for publication: J.T., D.Z.-A., D.S, H.M, C. vd V., A.H., I.V., P. vd B., F.V., L.G., S.A., J.F., D.L., R.F., T.R.

*Competing interests.* At least one of the (co-)authors is a member of the editorial board of Atmospheric Measurement Techniques.

*Disclaimer.* Judith Tettenborn was supported through a grant from Environmental Defense Fund.

*Acknowledgements.* We thank all who contributed to data acquisition during the measurement campaigns across various research groups. Special thanks to Roberto Paglini from Politecnico di Torino and Ceres Woolley-Maisch from Utrecht University for their dedicated efforts during the campaign, assistance with data analysis, and valuable discussions.


In the drafting and programming of this publication the AI tool ChatGPT (https://chat.openai.com/) has been utilized as aid based upon initial drafts. Every response stemming from the AI has been checked, evaluated and only implemented with care.



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
