# Peer review of "Improving Consistency in Methane Emission Quantification from the Natural Gas Distribution System across Measurement Devices"

_EGUsphere, 2024_

## Author Response (AR2)

Dear Dr. Dietrich G. Feist,

Thank you for your valuable suggestions. In the revised version we have separated regular text and mathematical formulae, and we changed superscript indices to subscript indices where they could be perceived as exponents. To further differentiate the mathematical indices (representing the Monte Carlo iteration, the release rate, and the number of transects) from the variable names (e.g., E for emission, "sim" for simulated), we introduced brackets around the variables.

Regarding your suggestion to replace the full expression of the logarithm of the $CH_4$ mole fractions with a variable name (e.g., Gamma), we would prefer to retain the explicit expression. We believe that keeping the mathematical expressions as explicit as possible improves clarity and will help a wider range of users to employ this method. Also, this approach keeps the notation style more consistent with Weller et al. (2019), which is the key publication we are building upon.

Regarding the brackets in line 157: the brackets are correctly closed, after additional information on the variable for the mole fraction $c(t_{i+1})$; the bracket closes after the words "background level" in the sentence: "...the $CH_4$ mole fraction enhancement at time $t_{i+1}$ ($c(t_{i+1})$, in ppm, after subtraction of the $CH_4$ background level) was multiplied...".

We hope these corrections meet your approval and once again thank you very much for your helpful comments and your work.

Kind regards,

Judith

The article Improving Consistency in Methane Emission Quantification from the Natural Gas Distribution System across Measurement Devices uses several controlled release experiments to devise a better way of inferring fluxes from measurements in the case of gas leaks from the natural distribution system. By analyzing all the data, they show that using the area instead of the maximal amplitude of the signal allows a better coherence especially when using different measurement technologies. They also confirm the importance of repeating transects to reduce errors.

The article is generally well written, and the authors spend time examining the weaknesses of their method (testing the linearity hypotheses, discussing how precise is the method...).

I recommend publication after a few minor modifications.

I would suggest to push the appendix in the Supplementary, it is a bit confusing to have both. It would also be nice to harmonize a bit the description of the measurement procedures in the Supplementary material.

- The appendix has been integrated into the SI.
- The Measurement Procedure section has been removed due to redundancy and lack of clarity. Relevant details about the setup have been integrated into the main text, while the instrument description has been relocated to the SI under the newly renamed section Instrument Characteristics.

In the supplementary material: l229 and 235, the Figure seems to be out of place or missing comas, please check.

- corrected

In the main text, I would make clear that all data have been used to derive the equation and then the equation has been used to derive the fluxes for all the measurements. There is no training data (at least that I what I understood).

- In the methods two sentences were added/changed:
  The fitting was performed using the entire dataset, without separating it into training and testing data.
  For each valid plume transect (i) in the dataset used to derive the regression model, emission rates were estimated utilizing both estimation methods, applying the inverse of Equation 1 and Equation 3.

l37 replace plattform by platform

- corrected

l41 replace are by being

- corrected

l64 (after "environment") and 66 (after now) and at other places in the text, additional comas would be welcome to facilitate the reading of long sentences, please check

- corrected
- additional comas were added in other parts of the text to improve readability

l234 add an  in "as an empirical model"

- corrected

l376 replace But by However

- corrected

RC2: 'Comment on egusphere-2024-3620', Anonymous Referee #2, 06 Feb 2025

**General comments:**

This manuscript describes how the quantification methods for methane emissions from natural gas distribution systems (leaks) can be improved and give practical recommendations. The strength of the manuscript is the extended study, which incorporates measurements from a number of different methane instruments placed at different measurements sites (three in Central Europe, one in Canada) carried out for a duration of several days at each site by different research groups. Various statistical methods were applied to analyse the available data set as a whole. The outcome of the study - integrated spatial peak area is more robust to use than the maximum methane enhancement - agrees well with previous findings by Maazallahi et al. (2020). However, it would be important to also highlight more in detail what is new and different in the present study compared to e.g. the Maazallahi et al. study.

The manuscript is well-written and generally logical structured with some exceptions. The number of figures and tables in comparison to the text is appropriate. In general, the figures are of good quality. Sometimes the labelling is missing or could be improved. Proper credit is mostly given to related work.

Details on suggestions for improvements on the above mentioned topics are given in the specific comments below.

The paper addresses scientific questions relevant to the scope of AMT.

For the reasons mentioned above and below the paper is appropriate for publication in AMT after a minor revision described below.

**Specific comments:**

Are any wind measurements available at least for some of the sites or re-analyses data (London)? Incorporation of such data would be interesting and important to evaluate the dependency of the results on the current wind situation. If the measurements were not performed perpendicular to the wind direction (as mentioned in the manuscript for London), a major part of the plume can be missed. Please discuss this in more detail.

Among the datasets used in this study, reliable wind data were only available to us for the Toronto CRE, which are covered in Gillespie et al. (2023). We attempted to collect wind data during the Rotterdam and Utrecht II experiments, but they proved unsuitable for further analysis. In Rotterdam the analysis shows that the measurements do not represent wind conditions realistically because of local obstructions (which are of course common in cities). During the Utrecht II experiment, we encountered data logger issues which led to a loss of data. We agree that wind information may be useful but unfortunately cannot provide this here.

Extended discussion in 3 Results&Discussion:

A wide range of peak maxima and areas is observed for each release rate. This illustrates the shortcomings of the quantification of emissions for individual leak indications using this simple statistical method. This spread reflects the nature of turbulent dispersion of plumes. Environmental factors, in particular built environment and meteorology, e.g. convection, complex wind patterns within urban areas (back circulation, wind channelling, blockages), turbulence and diffusion, play a major role in determining the location, shape and CH4 mole fraction of the meandering CH4 plume (Carpentieri and Robins (2015), Cassiani et al. (2020), Ražnjevic et al. (2022)). Still, a greater CH4 release increases the likelihood of detecting higher CH4 mole fractions. This is the relation that is

captured in the proposed transfer equation. Yet, changing wind conditions or turbulence can still cause the main plume to become highly diluted or be transported away from the street, leading to smaller peak maxima and spatial peak areas. At the extremes, high CH4 mole fractions are measured in one transect and no peak at all in another one (Luetschwager et al. (2021)). Ideally, measurements should be conducted downwind of the emission source and perpendicular to the wind. This approach is commonly used when targeting individual emitters, such as oil and gas processing stations or farms (Hensen et al. (2006), Korben et al. (2022)). However, when measuring emissions from urban gas distribution networks, the built environment imposes constraints on positioning. Incorporating wind measurements from the survey vehicle could potentially improve emission estimates and should be explored in future research.

Information on the atmospheric stability is not available from the present data set, however it would be interesting to compare the influence of the timing of the measurements on the results. Recommend to divide all flights into morning / noon / late afternoon flights to find out at what time of the day the best measurements can be achieved. At noon it can happen that the main plume rises straight upward and only a minor part is measured. In the morning / late afternoon accumulation of emissions is possible. Please comment more on this.

We can only compare measurements with the same release rate during different parts of the day, and there are only a few examples for this, so the analysis is not conclusive. In Rotterdam the 40 L/min release was carried out around 11:30 and 14:00, with no significant difference in measured peak area distribution whereas for the 20 L/min release rate, the one performed at 13:30 returns a lower estimate than the one at 11:00. During the Utrecht II experiment, 4 L/min was released at the same location during three time periods (10:48–11:22, 11:38–11:48, and 16:19–17:07), with no detectable differences in the spatial peak area distributions. For the London experiments, measurements in the early afternoon would yield slightly smaller release rates than in the late afternoon. The other release rates were applied only once, thus we cannot draw clear conclusions, unfortunately. We have added a sentence recommending the investigation of diurnal effects in future experiments. We note that von Fischer et al (2017) analyzed the influence of wind previously and reported the following:

*"We found that variation in wind speed had little to no impact on estimated leak rate. This finding is based on analyses of car based and weather station observations of winds captured during controlled releases and measurements in cities"* and

*"Given the spatial and temporal scales used here, it is not that surprising that wind speed measured on the top of the vehicle or from a local weather station is not strongly correlated with concentration. These measurements appear to not capture the turbulent processes that drive transport close to the leak."*

Page 16, Sect. 3.5: very important subsection! Improve by adding recommendations on wind direction measurements (see next sentence) and by adding the results from your missing study (my recommendation, see above) on the optimal probing time of the day (in case any useful results come out of that study). Add that the wind direction can be measured by an anemometer on the roof of the car or by a simple flag (tissue) reached out of the car window. Such simple measurements can improve the methane quantifications considerable.

Due to the inconclusive analysis, we could not provide recommendations on the optimal probing time of day. We added a recommendation in Sect. 3.5 to measure wind as well.

**Minor comments and technical corrections:**

General minor comment: "London" and "London I" (same meaning) are mixed in the manuscript (same for "Utrecht" and "Utrecht I"). Homogenize throughout the manuscript.

- corrected

Page 1, line 10-12: Is this true for all sites? In Fig. 1 it is only shown for Rotterdam and Utrecht (does it include all data from those sites?). Toronto is not shown and London (all data?) is presented in Fig. B1.

We have revised the abstract as follows to avoid ambiguity: Based on controlled release experiments conducted with various different devices in four cities (London, Toronto, Rotterdam, and Utrecht), emission estimation methodologies were evaluated. Indeed, when different analyzers were measuring in the same vehicle and from the same air inlet, the integrated spatial peak area was found to be a more robust metric across different methane gas analyzer devices than the maximum methane enhancement.

Page 2, line 24: re-order given references chronologically (and also alphabetically if same year, check throughout manuscript)

- corrected

Page 2, line 34: "present" à represent

- corrected

Page 3, line 64: "environments complex" à environments with complex

- comma added to clarify the meaning: Especially within urban built environments, complex wind flow, re-circulation, and dispersion patterns play an important role.

Page 3, line 72-73: Maazallahi (2020) gave the same recommendation as in your study. What have you achieved in addition to the mentioned study?

- Maazallahi (2020) indeed provided indications for this effect, based on limited measurements in one city by only two instruments from the same manufacturer. This was the motivation of our present study where we expanded on this analysis by comparing various instruments from different manufacturers using measurements from different times in three different locations. This makes the finding more robust.
- Further, even when this was indicated before, there was no way the peak area could be linked to emission rate because the "transfer equation" was missing. We establish a new statistical model based on several controlled releases, using the spatial peak area instead of peak maximum to replace the Weller method.
- To make this clearer for the reader, we specified in the introduction that Maazallahi (2020) only compared two different instruments, and that we compare in total eight.

Page 3, line 81: After the headline "2 Methods", briefly explain what subsections the reader can expect next.

- Added: In this section, we first describe the controlled release experiments conducted to generate the dataset (Sect. 2.1). We then outline the data processing steps, including peak identification and calculation of the Spatial Peak Area (Sect. 2.2). Next, we present the approach for estimating emission rates from the measurements (Sect. 2.3). Finally, we

evaluate the performance of the quantification method (Sect. 2.4), considering both its categorization ability and the impact of the number of transects on estimation accuracy.

Page 4, line 85: "SI" à Supplement Information (SI)

- corrected

Page 4, line 85: "Table A1" à Table A1 in the Appendix A

- The Appendices were merged with the Supplementary Information following the suggestion of Referee 1. Therefore, this sentence was changed to: … is given in the SI, Table S2.

Page 4, Table 1 legend: "Controlled release experiment" à Controlled release experiment (CRE)

- corrected

Page 4, Table 1: "Inlet Level" à Inlet Height

- corrected

Page 4, Table 1: Was the release height always the same (height of gas bottle?)? Add to legend text or somewhere else.

- Added to Table 1 legend: In all experiments, the release height was set at ground level.

Page 4, Table 1: For London and Utrecht, add I and II (second already incorporated).

- added

Page 4, line 88: "1a" à 2

- corrected

Page 4, line 95: "2a" à 2

- corrected

Page 4, line 98: write out TNO the first time

- added

Page 5, line 103: "1b" à 3

- corrected

Page 5, line 107: write out IMAU the first time

- added

Page 5, line 108-117: Give here also the release rates for London and Toronto (as above for Utrecht & Rotterdam).

- added

Page 5: line 110, line 111, line 116: Not necessary to mention companies, since listed in Table 1.

- removed

Page 5, line 132: "vehicle the" à vehicle, the

- corrected

Page 5, line 133: "dataset peaks" à "dataset, peaks"

- corrected

Page 6, line 136: "UU car" à UUAQ

- corrected

Page 8: Remove the dot after some of the equations.

- corrected

Page 9, After the headline "3 Results and Discussion", briefly explain what subsections the reader can expect next.

- added

Page 9, Fig. 1: Avoid placing a figure at the beginning of a subsection.

- changed

Page 9, Fig.1: Are all data from Rotterdam and Utrecht included? Same in Fig. B1 for London (it might also be an option to add this figure to the main manuscript since it is illustrative)? Why are not the data from Toronto shown?

All data from Rotterdam and Utrecht I are included. For the London CRE, data from Day-1 and Day-2 are displayed. On Day-3 only one instrument was used. In Toronto two instruments were used on Day-1, however, they were not installed in the same vehicle, so the measurements cannot be directly compared with our analysis.

Page 9, line 206-207: Discuss why the peak maximum measured by G4302 is clearly underestimated by the other instruments.

Our hypothesis is that this is a combination of the high cell pressure, high flow rate and measurement frequency of the G4302 analyzer, which lead to a sharp and high peak because of low residence times in the analyzer cell. We do not completely understand why the peak maxima are lower for the TILDAS and MGA10 instruments which have a comparable residence time in the measurement cell. So this requires further research.

Page 10, line 209: "0.63" à not in agreement with values given in Fig. 1

- corrected

Page 10, line 210: "most" à all

- corrected

Page 10, line 216: "Figure B1" à Figure B1 in Appendix B

- changed to (SI, Figure S21)

Page 10, Fig. 2: add (a) and (b) to legend text.

- added

Page 11, line 225: At several places comparisons to Maazallahi / Gillespie are made. Discuss more details from these papers in your manuscript and point out what is different/new in your study.

Introduction extended by:

Maazallahi et al. (2020) noted that the integrated spatial peak area is much more consistent than the peak maximum between two different instruments operated in parallel during mobile measurement campaigns in Hamburg and Utrecht. Gillespie et al. (2023) conducted controlled release experiments and street-level measurements in Toronto, using two CH4 devices. By applying Gaussian plume inversions to both CH4 peak maximum and peak area, they concluded that emission rates should be estimated using either the peak area or, if relying on enhancement height, incorporating explicit modelling of the instrument response function to account for variations in cell residence time.

Results&Discussion extended by:

Gillespie et al. (2023) and Maazallahi et al. (2023) made similar observations when comparing different analyzers. In this study, we generalize these results by demonstrating the same effect across eight different instruments, enhancing the robustness of this conclusion.

Page 11, line 228: Are there any plans in future to install such monitoring instruments downwind of the main wind direction or are they always randomly installed? Or is it worth adding such recommendations in your conclusions (Sect. 3.5)?

This is likely a misunderstanding, we mean deployments on mobile vehicles, not fixed installations of sensors.

Page 11, line 235-236: Add "[CH4] area" and "re" to the sentence.

- added

Page 12, Fig. 3: What is the black line? Why is it missing in (b)?

The grey line displays the Weller equation (which is only based on peak maxima and not available for the peak area). The caption was slightly changed to enhance understandability: The Weller equation (grey line in (a)) is displayed as a comparison...

Page 12, Fig. 3: "London Day 2" à London I Day 2

- added

Page 12, Fig. 3: In figure legend replace London by London I and Utrecht by Utrecht I.

- added

Page 13, Fig. 4: Add (a) and (b) to the top of the figures instead of repeating it below the figures. Write out the different categories directly into the figure.

The header Area eq. and Weller eq. were removed at the top of the figure, while the caption below was kept, to keep it consistent with the previous figures. Categories were added into the figure.

Page 13, Fig. 4 legend: "true emission rate rE" à true emission rate rR

- to keep it consistent with the rest of the manuscript, the in-figure label was changed from rR to rE

Page 13, Fig. 4: "pool which" à pool in (%) for which

- changed to: ...lines visualize the amount of plumes (in %) from each category pool which the algorithm classifies...

Page 13, Fig. 4: to be better comparable to Fig. 3, switch the order of (a) and (b). Add after (a) "Based on Peak Maximum" (as in Fig. 3) and after (b) "Based on Spatial Peak Area".

- changed

Page 13, legend: write out "CRE"

- changed

Page 13, line 274: but also underestimated

- added

Page 13, line 275: "22%" à 21%

21% of category 3 are underestimated in category 2, 1% in category 1. In total "(22%) of category 4 […] peaks are underestimated into category 1 or 2".

Page 14, line 276-277: Check the numbers given in the text with the numbers in Fig 4.

In terms of repair prioritization, it is important to differentiate the higher emission categories with significant emissions (category 3 and 4) from the lower emission categories. Here, it was checked how many plume transects that are truly in category 3 or 4 end up categorized as 3 **or** 4, that means that they are differentiated from low emission peaks. For category 3 peaks 43% + 35% = 78% of the leak indications are categorized as either 3 or 4. For category 4 peaks 30% + 64% = 94% are categorized as either 3 or 4.

The incorrect number 93% was changed to 94%.

Page 14, line 279: Add "(Fig. 4b)" to the sentence

- added

Page 14, Fig. 5: Avoid placing a figure at the beginning of a subsection. Don´t repeat headlines above and below the figures.

- corrected

Page 14, line 282: "London" à London I

- corrected

Page 14, line 285: "considered an" à considered as an

- corrected

Page 15, line 299: "Figure 5a" à Figure 5

- We are specifically referring to 5a here. We have added a sentence to further clarify why (see my answer below).

Page 15, line 299: Briefly describe what is shown in Fig. 5a compared to Fig. 5b.

Added:

This effect has been demonstrated in Luetschwager et al. (2021) and is illustrated in Fig. 5a. There, the Monte Carlo mean of the absolute percentage deviation of emission estimation, based on N transects ($N \in [2, 10]$), from the calculated mean emission rate, based on all M transects measured for this release rate, is shown. The percentage deviation from the mean decreases drastically with an increasing number of transects, and converges to 0 for $N = M$ (not shown).

Still, if the offset is not too large, the error in emission estimation decreases with more transects. This is visualized in Fig. 5b, where the Monte Carlo mean of the absolute percentage deviation of emission estimation in respect to the true emission rate is shown.

Page 15, line 319: "to the mean" à to the mean (Fig. 5a)

- added

Page 15, line 321: "for N=2" à for N=2 (Fig. 5b)

- added

Page 16: Fig. 6: label the y-axes left and right

Categories and y-axis name were added.

Page 16, line 334: "account three" à account one (Fig. 6a), three

- added

Page 16, line 337: Move this sentence to follow-up directly after line 336.

- changed

Page 16, line 337: "on a single detection" à on one single transect (N=1),

- changed

Page 16, line 338: "are used" à are used (N=3)

- changed

Page 16, line 339: "six transects" à six transects (N=6)

- changed

Page 16, Line 342: "measurements" à transects

- changed

Page 17, line 349: "six transects" à six transects downwind of the leak.

- Changed to 'six transects downwind of the leak indication'

Page 17, line 350: "subtract it" à subtract them

- corrected

Page 17, line 358: "inverse Area eq." à inverse Area eq. (Eq.13)

- added

Page 17, line 371: "emission rate" à emission rate (rE)

- added

Page 17, line 372: "enhancement" à enhancement ([CH4]area)

- added

Page 18, line 381: "Including…" à Including three transects instead of one improved correct…..

- added

Page 18, line 382: "and to 62-82%..." à and to 62-82% when incorporating six transects instead of one transect

- added

Page 19, line 388: „London" à London I

- changed

Page 19, line 389: „Utrecht" à Utrecht I

- changed

Page 19, Table A1: Three times „London" à London I

- changed

Page 20: „Utrecht" à Utrecht I

- changed

Page 20: Has the impact of the different timing been investigated (Toronto)?

Page 22, Fig. B1 legend text: "London" à London I and give the day(s)

- added

Page 24-27: Add more details to some of the references (e.g. ISBN number, http://) given in the following lines: 411, 423, 424, 426, 458, 488, 506

- ISBN and http:// added

Page 26, line 495: improve spelling mistake

- corrected

**Supplement**

**Specific comments:**

Does the main manuscript contain references to everything shown in the supplement? Either refer to the supplement material in the main document or leave it out completely.

- The main manuscript contains references to all sections in the Supplementary Information.

Page 6, line 30: Why was the release done from the surface level here? Normally I assume the release height was the height of the cylinder? Please add the release height at each site into one of the available tables (or state that it is constant). How do these release heights agree with heights of detected leaks in the field?

A teflon tube connected to the cylinder outlet was used to release the methane at ground level. As most natural gas pipelines in urban areas are underground, emissions to the atmosphere are expected from ground level.

Page 6, line 51: In general, for all sites with car probing: When the cars with the instruments were driving along the streets, where there also other normal cars using these streets at the same time? If

yes, this can also distort the measurements. Please comment. For the routine leak detection surveys described in the main Sect. 3.5, you can also add that the measurements should be carried out with a sufficient distance to previously passed cars.

In the London experiment, the release took place on an airfield with no other cars present. All other release locations are in the normal built environment, but very busy locations were avoided for safety reasons, but the experiments did not affect local traffic. In the Toronto experiment, releases were conducted at a parking lot and an industrial harbor. In Utrecht, between university buildings at the campus of Utrecht University, in Rotterdam, in a residential area with modest traffic.

Assessing the impact of other vehicles on plume shape and methane measurements may be an interesting topic for future research. The presence of other cars is likely to distort the methane plume, though the extent of this effect remains uncertain. Ideally, measurements should be taken without following another vehicle too closely. However, in urban environments, this is not always feasible.

'If possible, maintain a sufficient distance from previously passed cars while conducting the measurements.' was added in Sect. 3.5.

Page 28: Here you present the Figs. S23-S24 and S25-S26. Also add a few sentences on the interpretation.

**Minor comments and technical corrections:**

Page 1, line 8: „comparison" à Comparison

- corrected

Page 2, line 20: "equation" à equations (Eqs. 1-2)

- added

Page 2: Refer in the text to Table S1

- added

Page 3, 5$^{th}$ column: „Cell V" à Cell V$_{norm}$

- changed

Page 3, Table S1: add precision of all instruments

- added

Page 4: „Utrecht" à Utrecht I

- changed

Page 4: „London" à London I

- changed

Page 5-7: Recommend to replace all text by a table.

The Measurement Procedure section has been removed due to redundancy and lack of clarity. Relevant details about the setup have been integrated into the main text, while the instrument description has been relocated to the SI under the newly renamed section Instrument Characteristics.

Page 9, Fig. S5: Is the day shown in Fig. S5 the same as in Fig. S4? Are the measurements shown for G4302 in both figures the same? The first impression is that it isn´t?

Figures S4 and S5 show measurements from the same day but collected by different vehicles. The G4302 and Mira Ultra instruments were transferred from the UUAQ car to the TNO truck after the lunch break. The figures have been updated accordingly: Fig. S4 now only includes data until 10:45, when the UUAQ car stopped, while Fig. S5 shows data from the G4302 and Mira Ultra starting at 11:05, when the TNO truck resumed measurements after the lunch break, now carrying four instruments.

Page 9, Fig. S6: "Utrecht" à Utrecht I

- changed

Page 10, Fig. S7: Just left of 14:10, the straight increase is most likely due to missing values. Don´t connect such time periods with a line.

- Corrected for S7 and S8, where values were missing.

Page 10, Fig. S8: Around 13:00 the red curve shows a strange behaviour with a plateau.

- Due to missing values, corrected in the figure.

Page 10, Fig. S8: "London" à London I

- changed

Page 12, Fig S10(d): Replace below and in figure "Utrecht" à Utrecht I

- Replaced in caption below, caption above figure were removed for all panels to avoid redundancy.

Page 13, line 112: Refer to Figs. S11-14 in the text.

- added

Page 13, line 113: Specify more in detail what you mean with "noise" and give examples.

- An explanation was added

Page 13, Fig. S11 legend: "London" à London I

- changed

Page 14, Fig. S12: all legends in figure and text "Utrecht" à Utrecht I, same for London in Fig. S13

- changed

Page 15, Fig. S14: left figure (a) has the headline Toronto Day 2, in the legend text a different order is given

- corrected

Page 16, line 156: "rtaes" à rates

- corrected

Page 17, line 170-171: Does this agree to the experience from other similar studies (give references)?

We are not aware of other studies that have evaluated methane leak rate distributions with these statistical tools. The references refer to other applications of these techniques.

Page 18, Table S3: "Utrecht" à Utrecht I

- changed

Page 19, Table S5: "London à London I

- changed

Page 20, line 192: "Utrecht" à Utrecht I

- changed

Page 20, line 199: Add that this is related to London I.

- added

Page 20, line 204: Add that this is related to London I.

- added

Page 21-25, Fig. S16-S20: For a better overview, give the name of the release (e.g. Rotterdam) once for every page as a headline of all sub-figures.

When increasing the vertical height of the figures by adding a header, they would either no longer fit on one page or need to be reduced in size, compromising readability. Instead, the release name was highlighted in bold in the figure caption to improve clarity.

Page 22, Fig. S17: "Utrecht"à Utrecht I

- changed

Page 24, Fig. S19: "London" à London I

- changed

Page 26, line 217: "Figure S22 and Figure S21" à Figs. S21-S22

- changed

Page 26, line 217: After referring to the figures, also add few sentences on the interpretation.

Added: This means that at certain locations, the statistical model may not perform well due to specific characteristics of the built environment. For example, narrow streets with tall buildings can either create tunnelling effects with high wind velocities or block the wind, resulting in very low velocities, depending on their orientation relative to the main wind direction and surrounding structures. Even at the same location, varying weather conditions on different days can influence the plume shape, leading to fluctuations in categorization success rates. For instance, while on Day1 of the London II controlled release experiment over 50 % of low-emission-rate peaks were correctly classified, only 20 % of peaks in the same emission category were correctly classified the following day.

Page 26, Fig. S21: labelling of y-axes is missing

- added

Page 28, line 228-229: mention Fig. S23b first at the end of the sentence

- corrected

Page 28, line 234-235: mention Fig. S24b first at the end of the sentence

- corrected

Page 34, line 256: Add more details to reference (e.g. ISBN number)

- added

Page 34, line 268: Add more details to reference (e.g. ISBN number)

- added